# LiDARGrid: Self-supervised 3D Opacity Grid from LiDAR for Scene Forecasting

**Chuanyu Pan** [*] [†]        **Aolin Xu** [*]

**Abstract:** Timely capturing the dense geometry of the surrounding scene with unlabeled LiDAR data is valuable but under-explored for mobile robotic applications. Its value comes from the huge amount of such unlabeled data. Current 3D scene perception methods however heavily rely on data annotations during training, to be able to tackle the moving objects in the scene during inference. In response, we present LiDARGrid, a 3D opacity grid representation that can be instantly generated from LiDAR points through a novel neural volume densification network. As the generated opacity grid can accurately capture the dense 3D geometry of the scene in real time, we also propose a scene forecasting network to predict the future opacity grids. Both the densification and forecasting networks are trained by self-supervision on NuScenes autonomous driving LiDAR dataset. The overall pipeline is evaluated by forecasting future point clouds from the forecast scenes. It notably outperforms state-of-the-art methods in point cloud forecasting in all performance metrics. Beyond scene forecasting, our opacity grid representation also excels in supporting tasks such as moving region detection and depth completion, as shown by experiments.

## 1 Introduction

Contemporary mobile robotics, e.g. autonomous driving systems, demand accurate and timely perception of the surrounding scene. Traditional solutions perform object detection and pose estimation from camera and/or LiDAR data, which heavily rely on the amount and quality of data annotation as well as sophisticated sensor fusion. As huge amounts of unlabeled camera and LiDAR data are accumulated each year, there is a growing interest in both industry and research communities to develop perception solutions that can be based on unlabeled data, while at the same time overcome the inherent sensor limitations, such as the depth ambiguity of camera and the sparse and unstructured nature of LiDAR data.

One direction of technology advancement that addresses the above challenge is the 3D occupancy grid representation of the scene [1, 2, 3, 4, 5, 6, 7, 8]. This method has exhibited promise in various aspects of mobile robots, encompassing 3D object detection, segmentation, and scene reconstruction, primarily due to their accuracy and computational efficiency in modeling the 3D environment. Compared with camera, LiDAR has unique advantages in accurate 3D perception and robustness to different lighting conditions. However, most of these works derive their 3D grid representations from cameras, leaving the exploration of grid representation from LiDAR data largely uncharted. Furthermore, we notice that very few of these works delve into scene forecasting, which takes a few historical frames and aims at predicting future frames using certain representations, except for e.g. [6]. However, in [6], the occupancy grid is constructed using supervised learning where the annotation of dynamic objects are needed. More literature review and discussions about related works on 3D occupancy grid construction and forecasting are relegated to Appendix B.

---

[*]Honda Research Institute, USA. Email: `panchuanyu45@gmail.com, aolin_xu@honda-ri.com`
[†]Work done during an internship at Honda Research Institute, USA, now at Meshy.

8th Conference on Robot Learning (CoRL 2024), Munich, Germany.

To fill in these research gaps, we propose LiDARGrid, a dense 3D opacity grid representation from LiDAR points. Motivated by optical models of volume rendering [9] and neural volume rendering, e.g. NeRF [10], we conceptualize our representation as a 3D grid, where each voxel represents the LiDAR opacity, or more precisely, the incremental likelihood of a ray to stop marching by hitting some particle in that voxel. This representation allows for distance rendering of LiDAR points, such that one can compute the distance from any position to an object's surface within the grid, thus reconstructing the scene geometry. The proposed 3D opacity grid is generated from LiDAR points by an autoencoder-based densification network, trained by matching the rendered LiDAR distance with the ground truth distance. Though the densified grid may be converted to a binary-valued *occupancy* grid by simple thresholding, we keep it as a continuous-valued grid, which can be used as a universal representation of the 3D scene for diverse perception and forecasting tasks. For this reason, we name our representation as the 3D *opacity* grid.

We explore the application of this representation in scene forecasting, where a forecasting network based on 3D convolution and UNET architecture takes historical grids as input and predicts future grids. Notably, both the densification network and the forecasting network are trained on unlabeled LiDAR points, adopting a self-supervised approach that can readily access to a vast pool of training data. The performance of scene forecasting is evaluated by the point cloud forecasting metrics. Khurana et al. [11] is the most recent work that predicts future point clouds with grid-based representation and known LiDAR pose. It shows much crisper results than previous point cloud forecasting works [12, 13] with model-free prediction and unknown LiDAR pose, partly due to the fact that the 3D grid better preserves scene geometry. Advancing along the direction of [11] by incorporating the densification network and the improved forecasting network, we achieve notable improvements in the point cloud forecasting metrics, thus pushing forward the state-of-the-art performance of this task. We also provide comprehensive ablation studies to evaluate each module and justify the key design choices toward the improved scene forecasting. Besides scene forecasting, we extend the utility of our 3D grid representation to diverse applications, including moving region detection and depth completion. In summary, our contributions include:

1. We propose a 3D opacity grid representation for mobile robot perception. The representation can accurately capture the 3D scene geometry and can be generated in real time, which are crucial for various perception and forecasting tasks. The representation is learned solely based on unlabeled LiDAR data.

2. We propose a 3D convolutional forecasting network for our opacity grid. It outperforms the state-of-the-art approaches on point cloud forecasting tasks in all performance metrics.

3. We provide a comprehensive ablation study to elucidate the efficacy of our method, along with demonstrations of its potential in extended applications, such as moving region detection and depth completion.

## 2 Method

### 2.1 3D Opacity Grid Representation

**Representation Formulation** Our representation of the scene is inspired by the optical model for volume rendering used in computer graphics [9, 10]. It is assumed that the 3D space is permeated by particles that can scatter light, and the density of the particles varies across the space. We define the representation as a 3D grid of dimension $H \times W \times L$ with $H, W, L$ being positive integers, scaled by a factor of $S \in \mathbb{R}$ to align with the real-world metric in meters. In other words, $S$ is the voxel size in meters. The origin of the real-world coordinate system is centered within this grid. The continuous value of a voxel at position $(h, w, l) \in \{1, \ldots, H\} \times \{1, \ldots, W\} \times \{1, \ldots, L\}$ in the grid, denoted as $\sigma(h, w, l)$, represents

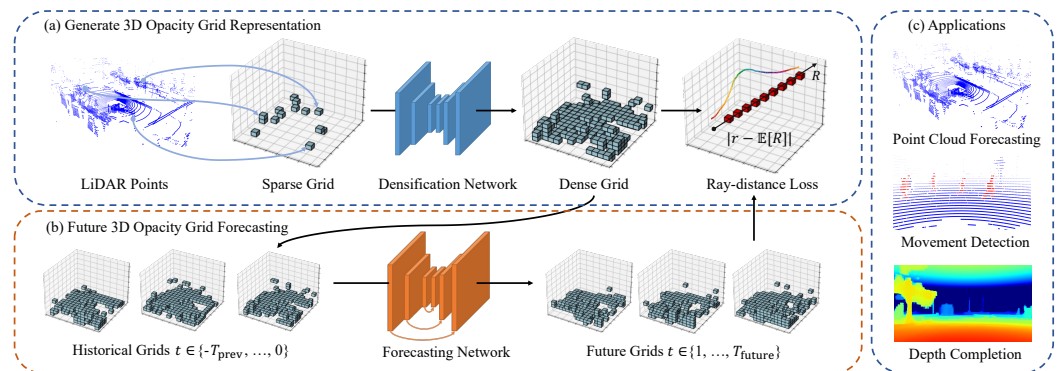

Figure 1: Overview of our approach. (a) To generate 3D opacity grid representations (LiDARGrid), our approach first initializes a sparse grid with LiDAR points and then transforms it into a dense 3D opacity grid through a densification network. Note that there are no skip connections in the densification network, unlike the forecasting network below. The training of the network is supervised by the ray-distance loss defined in section 2.1. (b) We illustrate the efficacy of our representation through scene forecasting, that is to predict a sequence of future grids from historical grids through a 3D convolutional forecasting network, which is also supervised by the ray-distance loss. (c) Utilizing our representation, we perform multiple tasks, such as point cloud forecasting, movement detection, and depth completion.

the density of particles at the real-world point $x_{hwl} = \left(S(w - \frac{W}{2}), S(l - \frac{L}{2}), S(h - \frac{H}{2})\right)$, which is approximately proportional to the number of particles within that voxel.

This representation is deterministic, but it gives rise to a simple probabilistic model of LiDAR point rendering: suppose the particles are uniformly placed at random within each voxel, and independently across voxels, then as the voxel size $S$ decreases to infinitesimal, $\sigma(h, w, l)$ becomes the rate of change of the probability of a LiDAR ray hitting any particle in that voxel [9, 14]. With a normal sized grid, given a ray $\mathbf{o} + \mathbf{d} \cdot r$ with origin $\mathbf{o}$ and direction $\mathbf{d}$, if the ray intersects with a sequence of voxels $(v_1, ..., v_n)$ at distance $(r_1, ..., r_n)$, the probability of the ray hitting a particle when traveling from $v_{i-1}$ to $v_i$ can be derived as [14]

$$\alpha_i = 1 - \exp\left(-\sigma(v_i) \cdot |r_i - r_{i-1}|\right). \tag{1}$$

We see that $\sigma$ essentially represents the opacity of each voxel: the larger the value $\sigma(v)$, the more likely that a ray stops at voxel $v$. Hence we name the proposed representation as *3D opacity grid*. Applying the above probabilistic point rendering model, the distance $R$ that a ray travels before stopping at a voxel is a random variable, taking value $r_i$ with probability

$$p_i = \alpha_i \cdot \prod_{s=1}^{i-1}(1 - \alpha_s). \tag{2}$$

The expected distance of a ray is then

$$\mathbb{E}[R] = \sum_{i=1}^{n} p_i \cdot r_i. \tag{3}$$

The process of tracing these rays can be likened to simulating LiDAR beams as they traverse through open space and terminate upon encountering an object's surface. Utilizing Eq. 3, given the 3D opacity grid, we can compute the expected distance between the starting point and the end point of a LiDAR ray.

**Initialize Grid with LiDAR Points**   To acquire the 3D opacity grid from LiDAR points, we first initialize a grid by mapping the LiDAR points to the grid. For a LiDAR point $(x, y, z)$ represented in the real-world coordinate, we map it to a voxel grid at position $\left(\lfloor \frac{z}{S} + \frac{H}{2} \rfloor, \lfloor \frac{x}{S} + \frac{W}{2} \rfloor, \lfloor \frac{y}{S} + \frac{L}{2} \rfloor\right)$ and set the voxel value to a constant $\sigma_0$. Applying this process to all LiDAR points, we can initialize a sparse grid of spatial occupancy.

**Volume Densification**  The major challenge we address is generating a dense and continuous-valued opacity grid from the sparse initial grid. A naive way would be applying a low-pass filter to the sparse grid, however its rigidity hinders the geometric consistency of the filtered grid. With this in mind, we design a learnable autoencoder densifier:

$$\mathbf{V}_{\text{dense}} = \mathcal{D}(\mathcal{E}(\mathbf{V}_{\text{sparse}})) \quad \mathbf{V}_{\text{sparse}}, \mathbf{V}_{\text{dense}} \in \mathbb{R}^{H \times W \times L} \tag{4}$$

The encoder $\mathcal{E}(\cdot)$ maps the sparsely initialized grid into a low-dimensional representation with a series of convolution layers, extracting low-frequency information from the input. The decoder $\mathcal{D}(\cdot)$ upsamples the intermediate features with transposed convolution to reconstruct a grid with the same dimension as the input. Importantly, unlike the popular UNET architecture, we do not include skip connections between the encoder and decoder layers in order to retain more low-frequency information when passing through the network. Experiments show that this design choice is crucial for achieving a good densification performance. The network is trained in a self-supervised manner with the ray-distance loss:

$$\mathcal{L}(P, \mathbf{V}_{\text{dense}}) = \sum_{\mathbf{o}+\mathbf{d}\cdot r \in P} \left| r - \mathbb{E}[R] \right| \tag{5}$$

where $P$ is a set of LiDAR points, $r$ is the ground-truth distance of a point to the LiDAR origin $\mathbf{o}$. $\mathbb{E}[R]$ can be calculated by Eq. 3 and is differentiable for backpropagation. We randomly rotate and translate the point cloud as data augmentation to acquire robust densification. The network eventually learns a pattern to map $\mathbf{V}_{\text{sparse}}$ to a dense representation $\mathbf{V}_{\text{dense}}$. Furthermore, the small-size intermediate feature vector can be stored as a compressed version of the 3D grid, which largely improves the memory efficiency of this representation, and can potentially be even more memory-efficient than storing the LiDAR points of a scene.

## 2.2   Scene Forecasting with 3D opacity grid

We apply our 3D opacity grid representation to scene forecasting and show that with accurate 3D geometry reconstructed, we are able to reach better performance on such tasks.

**Problem Definition**  The goal of scene forecasting is to predict how the surrounding scene evolves in the future. Encoding the scene with our grid representation, the problem can be defined as finding a forecasting model $\mathcal{F}(\cdot)$, so that

$$(\mathbf{V}_{t+1}, ..., \mathbf{V}_{t+T_{\text{future}}}) = \mathcal{F}(\mathbf{V}_{t-T_{\text{prev}}+1}, ..., \mathbf{V}_t) \tag{6}$$

where $\mathbf{V}_i \in \mathbb{R}^{H \times W \times L}$; $T_{\text{prev}}$ and $T_{\text{future}}$ are the numbers of history frames and future frames respectively. Following the setting of [11], for each frame, we first transform the LiDAR points from their local sensor coordinate to the coordinate at frame $t$ based on the knowledge of LiDAR pose in each frame. Then we use the transformed points to initialize and densify the grid as stated in Section 2.1. Note that this coordinate transformation is optional, but would be a reasonable way of using the proposed method in practice.

**Scene Forecasting Network**  Enlightened by the success of 2D convolution on image perception, we regard our 3D grid representation as a natural extension of the 2D occupancy grid and apply a 3D convolution network to predict future grids. It can preserve local spatial information while extracting important features due to its 3D shift equivariance. Our scene forecasting network $\mathcal{F}(\cdot)$ is a UNET-style [15] 3D convolutional encoder-decoder network. Each pair of corresponding layers in the encoder and decoder with the same feature size are connected by a skip layer. Following Eq. 6, the input $\mathbf{D}_{\text{prev}} \in \mathbb{R}^{T_{\text{prev}} \times H \times W \times L}$ and output $\mathbf{D}_{\text{future}} \in \mathbb{R}^{T_{\text{future}} \times H \times W \times L}$ are series of temporal-continuous frames. Note that it is a general framework that allows various designs on the intermediate feature layer. For example, one can add LSTM modules [16] to process the latent features. As a baseline model, we are using identity mapping for simplicity.

**Loss Function**  An intuitive way to train is to use $\mathbf{D}_{\text{future}} = (\mathbf{V}_{t+1}, ..., \mathbf{V}_{t+T_{\text{future}}})$ as supervision, where $\mathbf{V}_i$ is generated from the future ground-truth LiDAR points:

$$\mathcal{L}_{\text{voxel}} = \sum \left| \mathbf{D}_{\text{future}} - \hat{\mathbf{D}}_{\text{future}} \right| \tag{7}$$

However, due to high dimensionality, we observe that the network tends to learn an average result with this loss function because occupancy change on an object's surface barely affects the overall loss. Therefore, we propose to use the sparse ground-truth LiDAR points directly as a weaker supervision:

$$\mathcal{L}_{\text{ray}} = \sum_{i=1}^{T_{\text{future}}} \mathcal{L}(P_{t+i}, \hat{\mathbf{V}}_{t+i}) \tag{8}$$

where $\mathcal{L}(P_{t+i}, \hat{\mathbf{V}}_{t+i})$ shares the same definition in Eq. 5, $P_{t+i}$ is the set of LiDAR points at frame $t+i$. We compare these two loss functions in the experiment section.

**Point Cloud Forecasting**   Our method can be easily applied to point cloud forecasting tasks. Given future 3D grids $(\mathbf{V}_{t+1}, ..., \mathbf{V}_{t+T_{\text{future}}})$ and LiDAR rays $(\mathbf{o}, \{\mathbf{d}_i\})$, one can simulate future LiDAR points with volume rendering (Eq. 1 to Eq. 3). We follow [11] and set $(\mathbf{o}, \{\mathbf{d}_i\})$ from ground-truth future LiDAR rays.

## 2.3   Applications beyond Scene Forecasting

In addition to scene or point cloud forecasting, our 3D opacity grid representation can be applied to other tasks. We describe two additional tasks: moving region detection and depth completion in the experiment section, and show promising results on both of them.

## 3   Experiments

### 3.1   Setup

**Dataset**   Our experiments were mainly performed on the NuScenes dataset [17], which is a public autonomous driving dataset containing 1000 driving sequences collected by 6 cameras, 1 LiDAR, and 5 RaDAR sensors. We only use LiDAR data, which has a frame rate of 2Hz with around 20000 points per frame. We follow the NuScenes' setting and split the dataset into 850 training scenes and 150 testing scenes. We train our densification network and forecasting network on the training set and show evaluation results on the testing set.

**Grid and Networks Details**   Following [11], the grid covers a $4.5m \times 70m \times 70m$ 3D space. Specifically, we set the grid dimension and size to $H = 45$, $W = 700$, $L = 700$, and $S = 0.1$. Each voxel in the opacity grid is initialized with $\sigma_0 = 1$ if it contains a LiDAR point and 0 otherwise. Our densification network contains an encoder with 4 downsampling layers and a decoder with 4 upsampling layers. Each downsampling layer shrinks the grid size by 2 and doubles the channel size. Upsampling layers reverse this process. Our forecasting network shares a similar structure with 3D convolution and skip connections between each pair of encoder-decoder layers.

**Evaluation Metrics**   To ensure a rigorous comparison, we adopt the identical evaluation metrics as presented in [11]. Specifically, we report the in-grid error (L1) and relative in-grid error (AbsRel), measuring the accuracy of distance predictions along LiDAR rays. Additionally, we employ the vanilla Chamfer distance (Vanilla CD) and in-grid Chamfer distance (In-grid CD) [11, 12] to gauge the spatial distribution error of the predicted point cloud, with measurements expressed in square meters ($m^2$).

### 3.2   Evaluate Scene Forecasting with Point Cloud

**Baselines**   We benchmarked our method against the most recent state-of-the-art approaches in point cloud forecasting, including S2Net[12], SPFNet[13], and 4DOcc[11]. Notably, both S2Net and SPFNet utilize the range map[18] as the scene representation, translating the 3D point cloud into 2D and subsequently employing 2D convolution techniques. 4DOcc employs a similar grid representation but distinguishes itself by using sparse grids acquired from LiDAR points as input, and further processing them through 2D convolution.

| Method | Horizon | L1(m)↓ | AbsRel(%)↓ | In-grid CD↓ | Vanilla CD↓ |
|---|---|---|---|---|---|
| S2Net[12] | 1s | 3.49 | 28.38 | 1.70 | 2.75 |
| | 3s | 4.78 | 30.15 | 2.06 | 3.47 |
| SPFNet[13] | 1s | 4.58 | 34.87 | 2.24 | 4.17 |
| | 3s | 5.11 | 32.74 | 2.50 | 4.14 |
| 4DOcc[11] | 1s | 1.40 | 10.37 | 1.41 | 2.81 |
| | 3s | 1.71 | 13.48 | 1.40 | 4.31 |
| Ours | 1s | **1.12** | **9.04** | **0.68** | **1.74** |
| | 3s | **1.50** | **12.04** | **1.04** | **2.13** |

Table 1: Comparison with the state-of-the-art point cloud forecasting methods on NuScenes Dataset[17]. We follow the evaluation metrics in 4DOcc[11]. The 1s and 3s horizons refer to 2 frames and 6 frames respectively, for both input and output.

| Method | Horizon | L1(m)↓ | AbsRel(%)↓ | In-grid CD↓ | Vanilla CD↓ |
|---|---|---|---|---|---|
| w/o VD | 1s | 1.23 | 9.16 | 0.76 | 1.88 |
| | 3s | 1.52 | 12.53 | 1.07 | 2.18 |
| w/ VD | 1s | 1.12 | 9.04 | 0.68 | 1.74 |
| | 3s | 1.50 | 12.04 | 1.04 | 2.13 |

Table 2: Evaluation on point cloud forecasting without and with Volume Densification (VD).

**Quantitative Results** Table 1 presents a comprehensive comparative analysis between our method and the baseline models. Notably, our approach excels in reducing the average L1 distance error, showcasing an improvement of 0.28m and 0.21m for the 1s and 3s settings, respectively, when contrasted with the current state-of-the-art. Most notably, our method demonstrates a substantial reduction in the in-grid Chamfer distance error, achieving a significant improvement of 50% and 29% for the 1s and 3s settings, and simultaneously lowering the vanilla Chamfer distance error by 38% and 50% for the same settings. This robust performance not only attests to our method's proficiency in predicting the scene's underlying geometry (L1 and AbsRel) but also underscores its capacity to predict uncorrelated samples (Chamfer Distance), such as point clouds. Our method achieves this remarkable improvement in Chamfer distance, despite not having employed it as direct training supervision, underscoring the accuracy and robustness of our approach in predicting future scene geometry.

**Run time** The proposed pipeline can run at 5 frames/second on a Quadro RTX 6000 GPU, being able to support real-time applications. Further improvements may be gained from network optimization and CUDA acceleration.

### 3.3 Ablation Study

Several key designs contribute to the improved scene forecasting, including volume densification, 3D convolutional forecasting network, and its loss function. To thoroughly evaluate the effect of these modules, we conduct a comprehensive ablation study on them.

**Opacity Grid Densification** We argue that inputting dense and geometry-preserved 3D representation to the forecasting network is essential to the forecasting performance. In this section, we train and test our forecasting network with or without volume densification. For the 'without' setting, we simply initialize the grid with LiDAR points introduced in Section 2.1 and input it into the forecasting network, the same as what is done in [11].

Table 2 shows the forecasting results with and without volume densification. Upon introducing volume densification, we observed consistent improvements across all evaluation metrics for both 1s and 3s settings. These improvements underscore the consistent effectiveness of the volume densification module in enhancing the forecasting task. It is noteworthy that volume densification offers benefits beyond scene forecasting. We illustrate this through visualizations

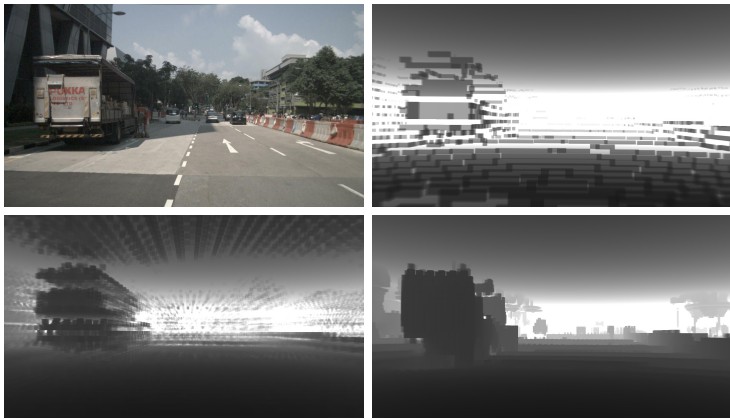

Figure 2: Visualizations of volume densification. Top-left: RGB image of the current view. Top-right: input sparse grid derived from LiDAR points. Bottom-left: a densification result from the network with skip connection. Bottom-right: a densification result from the network without skip connection, which is used in our method.

of the opacity grid, both before (top-right) and after (bottom-right) densification in Figure 2, revealing a densified 3D grid with fewer holes and a more continuous spatial distribution. This not only bolsters forecasting but also enhances its applicability to broader tasks, as discussed in Section 3.4, such as moving object detection.

**Densification Network**  We state in section 2.1 that the encoder of the densification network should be a low-pass filter. Since the supervision is sparse, letting high-frequency signals pass through the network will cause overfitting. We will show this by comparing the network with or without skip connections. Specifically, for the network with skip connection:

$$\mathbf{V}_{\text{dense}} = \mathcal{D}(\mathcal{E}(\mathbf{V}_{\text{sparse}})) + \mathcal{M}(\mathbf{V}_{\text{sparse}}) \tag{9}$$

where $\mathcal{M}(\cdot)$ is a linear mapping, others share the same definition with Eq.(4).

Figure 2 shows that the network with a skip connection generates grids with holes, where the occupied voxels are distributed discretely since the network is overfitting the sparse input. The network without a skip connection maps the input to a low-dimensional manifold and generates a continuous and dense grid after decoding.

**Loss Function**  In section 2.2, we introduced two distinct loss functions, namely $\mathcal{L}_{\text{voxel}}$ and $\mathcal{L}_{\text{ray}}$, employed in training the forecasting network. In this section, we delve into a comparative analysis of the performance outcomes associated with these two loss functions. In both scenarios, training is carried out until convergence is achieved. For the $\mathcal{L}_{\text{voxel}}$ setting, we apply the L1 loss to all voxels within the grid, utilizing the 'sum' reduction. The learning rate is set to 1e-6, with other conditions held constant.

Table 3 shows the results of applying these two loss functions. We show that compared to $\mathcal{L}_{\text{voxel}}$, using $\mathcal{L}_{\text{ray}}$ will drastically reduce the prediction error, on both L1 metric and chamfer distance. It is because, in areas obstructed from LiDAR rays, small noises are introduced due to the absence of information in the LiDAR-invisible regions. Employing $\mathcal{L}_{\text{voxel}}$ compels the model to accommodate these inaccuracies, leading to confusion and a tendency to neglect regions with accurate LiDAR supervision. Conversely, the use of $\mathcal{L}_{\text{ray}}$ mitigates this issue, resulting in a more accurate and robust forecasting model.

## 3.4  Extended Applications

**Moving Region Detection**  Through the alignment of two temporally adjacent 3D opacity grids within the same coordinate system, our method enables the detection of moving regions

| Method | Horizon | L1(m)↓ | AbsRel(%)↓ | In-grid CD↓ | Vanilla CD↓ |
|--------|---------|--------|------------|-------------|-------------|
| $\mathcal{L}_{\text{voxel}}$ | 1s | 2.48 | 23.27 | 1.53 | 2.45 |
|  | 3s | 3.92 | 41.70 | 3.22 | 4.27 |
| $\mathcal{L}_{\text{ray}}$ | 1s | 1.12 | 9.04 | 0.68 | 1.74 |
|  | 3s | 1.50 | 12.04 | 1.04 | 2.13 |

Table 3: Evaluation on point cloud forecasting by training with L1 Voxel Loss $\mathcal{L}_{\text{voxel}}$ and Ray-distance Loss $\mathcal{L}_{\text{ray}}$.

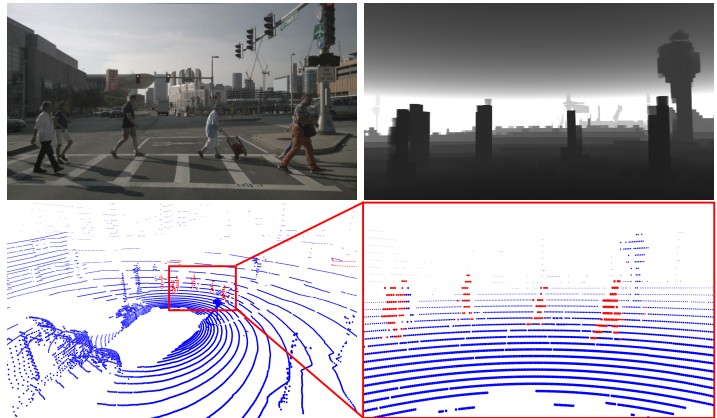

Figure 3: Moving region detection. Top-left: RGB image of current view. Top-right: rendered depth of current view within 3D opacity grid; darker colors indicating closer distances. Bottom-left: point cloud of entire scene; detected moving points are highlighted in red. Bottom-right: detailed view of detected moving points, which are pedestrians.

in the scene without relying on any object detection module. Traditional methods for this purpose e.g. [19] face challenges in effectively aligning LiDAR rays across consecutive frames due to the stochastic nature of ray directions over time. Our representation, as it can reconstruct dense scene geometry, overcomes this limitation. Figure 3 shows an example, where points on moving pedestrians are clearly detected by our method.

**Depth Completion** Existing camera-LiDAR-based depth completion require sophisticated fusion pipelines. Meanwhile, LiDAR alone offers only sparse depth maps. With our method, one can simply use the dense opacity grid generated from LiDAR to render depth values for any virtual camera pixel, following the principles outlined in Eq. 1 through 3. The promising performance of our method can be visualized in Figure 4.

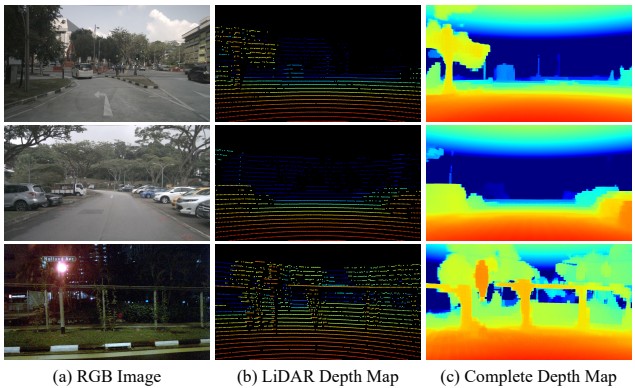

(a) RGB Image          (b) LiDAR Depth Map          (c) Complete Depth Map

Figure 4: Depth completion: dense depth map are solely from sparse LiDAR input.

## Acknowledgment

The authors would like to thank Seth Zhihao Zhao and Behzad Dariush for helpful discussions.

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

## A   Appendix: additional ablation studies

### A.1   2D Convolution vs. 3D Convolution

Some previous works [12, 16, 20] use 2D convolution in their forecasting network to process intermediate representation. 4DOcc [11] follows these works and applies 2D convolution to their 3D representation. We argue that 3D convolution will be a more suitable operation for our 3D grid representation than 2D convolution. Therefore, we conduct experiments to compare these two operations. For 2D convolution, we follow [11] and reshape the input from size $T_{\text{prev}} \times H \times W \times L$ to $(T_{\text{prev}} \cdot H) \times W \times L$.

Table 4 illuminates the forecasting results achieved by employing 2D convolution versus 3D convolution within the network. Notably, we observed a substantial improvement when transitioning from 2D to 3D convolution, particularly evident in the L1 and AbsRel metrics. This marked enhancement can be attributed to the intrinsic ability of the 3D convolution network to effectively extract 3D spatial information, courtesy of its equivariance property and synergy with the 3D grid representation. Furthermore, the 3D convolution framework affords the capability to process spatial and temporal information separately, thus providing a more versatile platform for the incorporation of modules designed to address temporal consistency—a testament to its broader utility and effectiveness in the forecasting task.

| Method | Horizon | L1(m)↓ | AbsRel(%)↓ | In-grid CD↓ | Vanilla CD↓ |
|--------|---------|--------|------------|-------------|-------------|
| 2D Conv | 1s | 1.30 | 10.42 | 0.79 | 1.91 |
|         | 3s | 1.81 | 15.00 | 1.39 | 2.53 |
| 3D Conv | 1s | 1.12 | 9.04 | 0.68 | 1.74 |
|         | 3s | 1.50 | 12.04 | 1.04 | 2.13 |

Table 4: Evaluation results on point cloud forecasting metrics with the forecasting network involving 2D and 3D convolution.

## B   Appendix: additional literature review

### B.1   3D occupancy grid scene representation

The 3D occupancy grid representation has gained increasing attention in recent years. Tesla introduced a real-time camera-based neural network solution for the 3D occupancy grid in 2022, named occupancy network [1]. The major advantage of such scene representation is that its accuracy need not be limited by the object-level annotation, thus enabling 3D perception of scenes containing unseen and irregular objects, either static or moving. Up-to-date open-sourced studies on occupancy network alternatives include Surroundocc [3], TPVformer [2], OccNet [21] and FBOcc [22] to name a few, all of which are camera-based.

Conventional LiDAR-based 3D occupancy representations mainly appear in the 3D mapping and segmentation literature, as reviewed in [23], and are far less studied compared to a large volume of literature on 2D occupancy from range sensors [24]. Most LiDAR-based 3D occupancy relies on approaches such as kernel methods and graphical models or other Bayesian methods to densify LiDAR point clouds, mostly resulting in a continuous volume representation. LiDAR-based 3D occupancy grid first appears in [11], however, the representation is solely constructed by placing point cloud in the grid, hence sparse. Our 3D opacity grid representation is directly tied to the volume rendering model in computer graphics [9, 10], where it is assumed that the space is permeated by particles, and the value of each voxel in the grid represents the density, proportional to the number, of particles in that voxel. The representation is constructed by densifying LiDAR point cloud using a carefully tailored neural network, hence dense, and it can run in real time.

Methods of obtaining ground-truth 3D occupancy grid from camera and LiDAR are discussed in [1, 25]. In these works, the moving objects are first detected and then separated from static backgrounds, so that the LiDAR point clouds on different types of objects can be identified and averaged out differently. In contrast, the training of the neural networks we use for opacity grid generation and forecasting does not rely on object annotation at all, and the trained densification network produces a dense opacity grid solely based on the LiDAR point cloud in the current frame.

Rigorously speaking, our 3D *opacity* grid is not equivalent to a 3D *occupancy* grid; instead, it is more of an intermediate and universal representation of the 3D scene. Nevertheless, as the density of particles in a voxel reflects the extent to which that voxel is occupied, an opacity grid may be converted to an occupancy grid by thresholding the value of each voxel into a binary value. The quantitative comparison between the 3D opacity grid of a scene generated by our method and its ground truth 3D occupancy grid is left as a future study. Some relevant theoretical and experimental investigations to this problem can be found in [26] and the reference therein.

Some recent works propose to generate 3D scene representations e.g. triplane, NeRF or Gaussian splatter, by directly feeding images to trained neural networks, including [27, 28, 29, 30, 31]. Conceptually our method is closely related to these works, in that we directly generate a 3D opacity grid by feeding LiDAR points to a densification network. An important feature of our method is that we explicitly use the NeRF-like volume rendering model to derive the ray-distance loss for training, which distinguishes our method from the pure model-free methods.

## B.2 Scene or occupancy grid forecasting

The problem of occupancy grid forecasting is to predict how the occupancy grid evolves in time. It arises when the scene contains moving objects. The majority of existing works toward this problem consider the prediction of 2D dynamic occupancy grid [32, 33, 34, 35, 36]. The prediction of a scene represented by a 3D occupancy grid appears to be a new problem, and is initiated by [11]. Different from conventional object-based scene prediction, the occupancy grid prediction under consideration is self-supervised, meaning that there is no need to perform object detection and tracking, and unannotated sensor data is directly used for prediction.

We have to distinguish the prediction problem studied in this work, which is about estimating *future* scene, from many existing works titled in terms of occupancy grid prediction, which are about scene completion or estimating occupancy of occluded areas. Works in the latter category include [37, 3, 22, 25]. To this end, we term the problem under study as opacity grid *forecasting*.

## B.3 Point cloud forecasting, 3D flow, and moving region detection

Point cloud forecasting has also arisen as a new problem in computer vision for robotic perception in recent years [12, 13, 11], as it has the potential to benefit downstream tasks with a large volume of unannotated LiDAR data. Earlier works on this problem [12, 13] do not separate sensor movement from the scene and predict the point cloud in a fully model-free manner. The recent work [11] reformulated this problem with a fixed reference frame, but leveraging known sensor pose, and improved the forecasting accuracy by leveraging the 3D occupancy grid representation. We follow the direction proposed in [11], and propose new methods for 3D opacity grid construction and forecasting to further improve the performance. Another closely related work is [38], where an implicit continuous occupancy field is learned without supervision for point cloud forecasting; in contrast, we utilize the discrete grid representation and explicitly forecast the grid. Some more recent works on point cloud forecasting also rely on camera-based object detection, e.g. [39].

Another advantage of working with LiDAR data is that the time evolution of the point cloud can provide information on object movement in 3D space. This problem has been studied in [40], where an end-to-end framework on point cloud flow is proposed. Some recent works on 3D occupancy completion and annotation also take the flow into consideration, by estimating the velocity in 3D of each voxel in the grid [1, 25]. There are also plenty of works on motion estimation from 2D dynamic occupancy grid, e.g. [33] for an end-to-end learning-based approach and the references therein. In those works, a challenging issue is the extreme imbalance between static and dynamic cells, and pixel-wise balancing is usually applied in the loss function counteracting. We also propose a solution for moving region detection based on our densified 3D opacity grid, where we synthesize the point cloud in the previous scene but in the ray directions of the current scene, and detect the distance change of the point cloud which gives hints on the a moving region.

