# OpenReview forum: "LiDARGrid: Self-supervised 3D Opacity Grid from LiDAR for Scene Forecasting"
_robot-learning.org/CoRL/2024/Conference — CoRL 2024_

### Official Review · Reviewer_rLH6 · 2024-07-18
**Weak accept with further questions.**

**Originality:** 3
**Technical Quality:** 3
**Clarity Of Presentation:** 3
**Potential Impact:** 3
**Recommendation:** 3
**Confidence:** 4

**Review:**

This paper presents a new 3D opacity grid representation derived from LiDAR points with neural volume densification and differentiable volume rendering to support tasks like scene forecasting, moving region detection, and depth completion. This innovative approach merges the concepts of neural radiance fields and explicit LiDAR point clouds to provide a novel form of 3D neural representation.

### Strengths:
* The 3D opacity grid combines the idea of neural radiance fields and explicit LiDAR point clouds, brings a new paradigm of 3D neural representation.
* The new representation is tested on 3D scene forecasting task and achieved superior performance.
* The presentation of this paper is generally clear and easy to follow.

### Weaknesses:
* Authors provided qualitative showcases of additional tasks like moving region detection and depth completion. It would be better to also have quantitative comparisons.
* Explicit `Limitations` section would be great.
* Some additional clarification and questions need to be answered.

**Quality Of The Limitations Section:**

1

**Questions For Rebuttal:**

* Is it possible to move the `related works` section to the main paper?
* In the self-supervised training of the densification network, does all the test points in `ray-distance loss` also inputed to the encoder? If so, will this harm the generalization ability? More analysis is appreciated.
* In the line `155`, what exactly is the `identity mapping`? need more details.

**Robotics Focus:**

3

**Summary Of Paper:**

This paper presents a new 3D opacity grid representation derived from LiDAR points with neural volume densification and differentiable volume rendering to support tasks like scene forecasting.

**Summary Of Recommendation:**

Generally acceptable but need further clarification and information.

---

### Official Review · Reviewer_V6p6 · 2024-07-19
**LiDARGrid Review: Novelty and Experiment concerns.**

**Originality:** 3
**Technical Quality:** 2
**Clarity Of Presentation:** 3
**Potential Impact:** 2
**Recommendation:** 3
**Confidence:** 4

**Review:**

The paper's strength is the novel viewpoint with respect to what voxel grids with density values actually are.

Unfortunately, while viewing a density value as proportional to opacity in interesting, it is unclear how this redefinition of the representation leads to a novel method / improved performance. The authors make the difference to an occupancy grid clear, however the difference with respect to papers that provide values for each voxel - whether density, features created by a small pointnet applied to the points in the voxel, etc, is unclear.

As a direct example, 3D Semantic Segmentation with Submanifold Sparse Convolutional Networks (by Benjamin Graham; Martin Engelcke; Laurens van der Maaten) was a CVPR paper in 2018 which also converts point clouds to a voxel grid, and creates a density-based value for each voxel before applying a 3D Unet. However there are countless others which also fit this (broad) description.

Specifically, their proposed method is:

1) Authors first take a standard voxel density grid and redefine it as an "opacity grid".
2) A learnable autoencoder is used to densify the grid, as a single LiDAR Scan results in a sparse scene.
3) Either a UNET-style 3D convolutional network predicts future grids, or predicts moving regions, or completes a sparse depth map.

None of these steps appear particularly novel, other than being applied to scene forecasting instead of an alternative 3D scene task.

**Quality Of The Limitations Section:**

1

**Questions For Rebuttal:**

There are many concerns, but the main issues from my perspective are that:

1) An opacity grid as described is just a voxel grid with each point having a density/opacity value. This is not a novel concept, with many existing works operating on voxel grids where the voxels are not simply binary "occupied or unoccupied".
2) The main contribution is scene forecasting, however the method to achieve this is simply a Unet style autoencoder with no notable technical novelty. While demonstrating that a voxel based representation outperforms e.g. 2D ones is interesting, that alone does not feel sufficient as a contribution.
3) For other proposed applications qualitative data is provided without either quantitative results or comparison to existing methods. While sufficient to demonstrate broad application potential, it does not demonstrate whether the proposed method is competitive for those applications.

There are also various minor areas to address, from occasional grammar mistakes, to the confusion created by figure 2 being coloured based on distance, despite being an "opacity grid". At first I was expecting it to be coloured based on opacity. Also unsure why the "sky" in figure 2 is dark when in theory it should be both completely opaque and have a distance of infinity.

But those minor areas are just areas to consider updating for any final edits, it's the main three which prevent me from being able to accept this paper for submission.

**Robotics Focus:**

3

**Summary Of Paper:**

Authors propose a combination of Unet-style 3D network and learnable autoencoder for various operations on voxel-grid scenes.

**Summary Of Recommendation:**

Method does not appear novel, nor are experiments extensive enough to accept as an in-depth analysis/ablation of applying 3D convolution Unet to various 3D scene problems.

---

### Official Review · Reviewer_8n74 · 2024-07-19
**Good paper on point cloud forecasting**

**Originality:** 3
**Technical Quality:** 4
**Clarity Of Presentation:** 3
**Potential Impact:** 3
**Recommendation:** 3
**Confidence:** 4

**Review:**

Strengths:
- The paper addresses an important open problem
- The proposed method is conceptually simple yet appears to be effective
- Strong results that improve SOTA of the nuScenes lidar forecasting task
- The paper is well written and easy to follow, with a few gaps discussed below

Weaknesses:
- There is no related work section. Some related work is mentioned in the introduction but not to a sufficient depth. In particular it would be important to better understand how exactly the proposed method differs from the baselines (SPFNet, 4dOcc); what other pointcloud representations have been proposed before; and discuss pointcloud forecasting with different input modalities.
- There is no conclusion section, and limitations are not discussed.
- It is unclear to me if the baselines in Table 1 represent the SOTA, it would be useful to justify the choice of baselines, and discuss if other better performing methods exist (potentially with different inputs / non-grid-based representation). For example, VIDAR [1] has demonstrated better results than 4dOcc with visual input.
- Ablation results in Table 2 reveal that the contribution of learned dense voxel representation is not substantial (SOTA is achieved even without volume densification). I wonder where does the performance gain come from then?

Minor:
- In line 131, is the learned representation indeed smaller to store than to original point cloud? This would depend on the size of reduced voxel, and the number of feature dimensions.
- it would be good to elaborate on what makes the learned voxel "dense" if it is only supervised with the same sparse point cloud? I wonder if randomly masking out parts of the input pointcloud (i.e. using a masked autoencoder) could further help the network to learn a densified representation
- it would be helpful to explain the metrics in more details

[1] Yang, Zetong, et al. "Visual point cloud forecasting enables scalable autonomous driving." Proceedings of the IEEE/CVF Conference on Computer Vision and Pattern Recognition. 2024.

**Quality Of The Limitations Section:**

1

**Questions For Rebuttal:**

See weaknessess listed above.

**Robotics Focus:**

3

**Summary Of Paper:**

A self-supervised lidar pointcloud forecasting method with a learned 3d opacity voxel representation of lidar point clouds, and a convolutional forward prediction network. The method achieves SOTA on nuScenes lidar forecasting task.

**Summary Of Recommendation:**

Good paper on point cloud forecasting, i recommend to accept given that the current weaknesses can be addressed.

---

### Author Rebuttal · Authors · 2024-08-09

We sincerely appreciate the valuable comments from all reviewers.
Common responses to all reviewers:
1. A detailed Related Work was relegated to Appendix B in the original submission. It reviews on (B.1) 3D occupancy grid scene representation, (B.2) scene or occupancy grid forecasting, and (B.3) point cloud forecasting, 3D flow, and moving region detection.
2. A Conclusion was not included to save space, as we attempted to write the Introduction section in a way to serve for the purpose of conclusion as well.
3. We discussed the limitation of the proposed method in Introduction and in Section 2.1: the main purpose of our NeRF-inspired opacity grid representation is for lidar point forecasting, not for reconstructing the dense occupancy grid per se. Though the opacity grid may be converted to a binary occupancy grid by thresholding, the resulting grid may not be the same as the true occupancy grid, which is usually estimated by dense lidar scans + camera inputs + supervised object detection by many recent methods. This is also why we did not compare them directly. Nevertheless, from sota-advancing lidar forecasting results, and qualitative depth estimation and moving region detection results, we see the high quality 3D geometry capturing capability of our representation.
4. The reason for the high quality 3D geometry capturing capability of our representation comes from not only the autoencoder densification network structure, but also the optical model underlying NeRF and the differentiable ray-distance loss function used in training.

Individual responses are in official comments.

---

### Decision · Program_Chairs · 2024-09-04

**Decision:**

Accept

**Comment:**

This paper proposes a 3D opacity grid representation learning approach for LiDAR points for scene forecasting. The reviewers find the paper generally well written and to be addressing an important problem. They appreciate the simplicity of the approach and the good results. However, the reviewers have also raised several concerns including lacking comparisons with state-of-the-art methods, concerns on the effectiveness of the dense voxel representation, unclear relation to certain existing work which raises the question of novelty, experiments lacking in-depth analysis, lacking quantitative comparisons, among others. Importantly, related work. conclusions, and limitations sections are missing. Each of the reviewers has listed several questions to be addressed in the rebuttal.
Post-rebuttal: Most of the reviewers' concerns have been sufficiently addressed. Please address the remaining suggestions in the final version.